# Complication Rates after Breast Surgery with the Motiva Smooth Silk Surface Silicone Gel Implants—A Systematic Review and Meta-Analysis

**DOI:** 10.3390/jcm12051881

**Published:** 2023-02-27

**Authors:** Marie-Luise Aitzetmüller-Klietz, Siling Yang, Philipp Wiebringhaus, Sascha Wellenbrock, Mahmut Öztürk, Maximilian Kückelhaus, Tobias Hirsch, Matthias Michael Aitzetmüller-Klietz

**Affiliations:** 1Division of Plastic Surgery, University Hospital Muenster, Waldeyerstrasse 1, 48149 Muenster, Germany; 2Department for Plastic and Reconstructive Surgery, Institute for Musculoskeletal Medicine, Westfaelische Wilhelms-University Muenster, 48157 Muenster, Germany; 3Department of Plastic, Reconstructive, and Aesthetic Surgery, Hand Surgery, Fachklinik Hornheide, Dorbaumstrasse 300, 48157 Muenster, Germany

**Keywords:** breast implants, breast surgery, nanosurface, breast augmentation, breast reconstructions

## Abstract

Background: In an era where textured devices are being phased out due to concerns about BIA-ALCL, the Motiva SilkSurface breast implants intend to alleviate historical prosthesis-related complications. However, its safety and feasibility remain unelucidated. Methods: An analysis of Pubmed, Web of Science, Ovid, and Embase databases was performed. A total of 114 studies were identified initially, and 13 of these met the inclusion criteria and were assessed regarding postoperative parameters such as complication rate or follow-up period. Results: In 4784 patients who underwent breast augmentation with Motiva SilkSurface breast implants, a total of 250 (5.2%) complications were observed. Short- and medium-term complication rates ranged from 2.8–14.4% and 0.32–16.67%, respectively. The most common complication was early seroma (*n* = 52, overall incidence = 1.08%), followed by early hematoma (*n* = 28, overall incidence = 0.54%). The incidence of capsule contracture was 0.54% and breast implant-associated-anaplastic large cell lymphoma was not observed. Discussion: Although the majority of the studies in the current literature suggest the distinction of the Motiva SilkSurface breast implants in terms of postoperative complications and capsular contracture, its safety and feasibility need to be further elucidated with well-designed, large-scale, multicenter, prospective case-control studies. Other: No funding was received.

## 1. Introduction

According to the annual report of the American Society of Plastic Surgeons (ASPS), breast augmentation has been one of the five most frequently performed cosmetic procedures from 2006 to this date. Alone in 2020, 193,073 patients underwent breast augmentation, of whom 103,485 (54%) received breast implants [1]. Despite the paucity of global statistics, the number of patients with breast implants worldwide is estimated to be 35 million [2]. Considering the size of this patient population and the disconcerting previous experience with specific implants, the surveillance of these devices cannot be overstated.

Historically, breast implants have undergone at least three accountability crises since their invention in 1960 [3], resulting in considerable negative physical or psychological impact on those patients involved. In addition to the detrimental repercussions of the 1992 Dow Corning crisis and the 2010 Poly Implant Prothesis (PIP) crisis [3], breast implant-associated anaplastic large cell lymphoma (BIA-ALCL), which was initially described in 1997 [4], emerged as the third crisis of the last decades. In breast plastic surgery and especially in oncoplastic breast reconstructive surgery, acellular dermal matrices are used, which significantly improve surgical conditions by also promoting healing and forming aesthetic scars [5].

In an era where textured surface breast implants are globally recalled, Establishment Labs (Alajuela, Costa Rica) launched the SmoothSilk^®^/SilkSurface^®^ technology. This led to the innovation of Motiva SilkSurface, which became the pioneer of the sixth generation of silicone breast implants. Since its introduction to the market in 2010, Motiva implants have undergone four major modifications (Figure 1). The first generation introduced the concept of SilkSurface to the market and remains one of the most favored breast implants to this day. Two years later, in addition to the enhancements in silicone gel (ProgressiveGel^®^PLUS), a new barrier technology (Bluseal^®^) was introduced to preclude potential silicone leakage. In 2014, Establishment Labs released their third-generation implant—Motiva Ergonomix^®^, which was featured to have “The most Natural Look and Feel” thanks to their newly utilized TrueTissue Technology^®^. By combining the unique properties of the ProgressiveGel Ultima^®^ and special elastic elastomer shell, they were able to mimic the natural breast tissue by allowing the downward shifting of the point of maximum projection in an upright position to form a natural teardrop shape. Aside from the peculiarities of the previous version, the latest Motiva implants, Motiva Ergonomix2^®^, embodies enhanced high-strength silicone dispersion, also known as the Motiva SuperSilicones^®^. It is suggested to have better mechanical properties as well as better adaptation to changes [6,7,8].

For the assurance of product traceability, Establishment Labs also retails its implants with Q INSIDE SAFETY TECHNOLOGY™ where a radio-frequency identification device (RFID) is embedded in the prosthesis and transmits accurate product information when read with an RFID reader. Despite the promising advantages, the adoption of RFID has been subject to controversy due to artifacts in the screening of high-risk cancer patients and the potential violation of patient confidentiality.

Although the company claims that the Motiva implants represent the most innovative devices available today, the clinical evidence to demonstrate its safety and superiority is limited. This study investigates the outcome and complication rates of Motiva SilkSurface breast implants in clinical use, to provide an evidence-based safety assessment.

## 2. Materials and Methods

### 2.1. Retrieval Method

The study was performed in accordance with the Cochrane handbook and PRISMA guidelines [9,10]. The literature search and retrieval as well as the data extraction was carried out collectively by two research fellows on 13 July 2022 using the Pubmed, Web of Science, Ovid, and Embase databases. The retrieval was conducted using the keywords “Motiva implant” without any limitations regarding language, time, or article type due to the limited number of publications in the literature. Randomized controlled trials, and cross-sectional and cohort studies on Motiva implants were included. Figure 2 demonstrates the structure of the current study. The follow-up periods were stratified into short-term (less than or equal to 1 year), mid-term (more than 1 year and less than 5 years), and long-term (more than or equal to 5 years).

### 2.2. Inclusion Criteria

The studies in which the patients underwent breast augmentation using Motiva SilkSurface implants;The study provided detailed raw data on surgical outcomes, i.e., postoperative complications;The number of patients included was greater than two.

### 2.3. Exclusion Criteria

The reviews, meta-analyses, conference reports, letters, expert consensus, and other types of literature in which the clinical data were not available;Case reports or case series with less than three patients included;The manuscripts which were not available in full text.

### 2.4. Acquisition of the Clinical Data

The literature screening and data acquisition were performed by two research fellows in our department data in line with the aforementioned inclusion and exclusion criteria. The acquired data included:Basic Characteristics such as the First Author, Date, and Country of the Publication, the Journal, Interval of the Study, Follow-Up Period, Number of Patients and Number of Prostheses, and the DisclosuresPatients’ Baseline Information: Age, Height, Weight, Body Mass Index (BMI); If no Information was Available this was Highlighted in the ManuscriptInformation on Breast Implant and Surgical Procedures: Name of the Implant, Breast Augmentation Type, Surgical Approach, Breast Implant Volume, and Profile/Projection as well as the Number of SurgeonsSurgical Outcome by Means of Complication Rates

### 2.5. Statistical Analysis

Statistical and graphical analyses were performed using SPSS Advanced Statistics Software version 22.0 (SPSS Inc., Chicago, IL, USA). Categorical variables are expressed using frequencies and percentages.

## 3. Results

The PRISMA flow chart of the study is depicted in Figure 2. Subsequent to the literature search using Pubmed, Embase, Web of Science, and Ovid databases, and removing the duplicates, 54 articles were listed for initial evaluation. A total of 39 articles were excluded (irrelevant literature (*n* = 10); did not meet the inclusion criteria (*n* = 17); conference reports, expert consensus, corrigendum, or discussion (*n* = 12)). The remaining 15 studies were then reevaluated for inclusion. Two of these studies included either patients with different types of implants or did not provide sufficient data to be included (Figure 2). This resulted in the inclusion of 13 papers [11,12,13,14,15,16,17,18,19,20,21,22,23,24,25], with a total number of 4784 patients.

The majority of articles were from Asia—specifically from Korea (54%, *n* = 7). The countries of origin for included studies are shown in Figure 3. Similarly, the distribution of the studies per continent can be seen in Figure 4. The follow-up ranged from 4.2 months to 72 months. In terms of study design, retrospective analyses (77%, *n* = 10) constituted the majority. Eight of the 13 studies (62%) received funding from the corresponding device manufacturer or disclosed personal financial interest for an author with the device company. The characteristics of the included studies and baseline patient information are summarized in Table 1.

### 3.1. Implanted Devices

The generation, volume, profile, and projection of the Motiva SilkSurface implants that were used in each study are shown in Table 2. Third-generation Motiva breast implants (Motiva Ergonomix™) were preferred most frequently. As implants with different sizes can be used in the same patient to achieve symmetry, we refrained from using the number of patients for calculations and instead used the number of implants.

### 3.2. Operative Data

The majority of the patients (89%) underwent primary breast augmentation. In terms of the surgical incision and pocket selection, inframammary fold (IMF), and dual-plane techniques were the most frequent approaches. More than half of the studies irrigated the pocket with antibiotic solutions to mitigate a postoperative infection. Montelukast sodium, which is considered to reduce capsule contracture, was utilized only in two studies. Detailed operative data are shown in Table 3.

### 3.3. Complications

Of the 4784 patients who received breast augmentation with Motiva SilkSurface breast implants, 251 patients faced complications with an overall incidence rate of 5.25% (Table 4). The incidence rates for individual complications can be seen in Figure 5. Excluding miscellaneous complications, marked as “Others”, the most common complication was early seroma (*n* = 52, overall incidence = 1.09%) which was followed by early hematoma (*n* = 28, overall incidence = 0.59%). Short-term complication rates ranged from 2.63–15.79% while medium-term rates ranged from 0.32–16.67% (Figure 6). A reoperation was needed in only five studies and the need for reoperation ranged from 0% to 8.57%. These were mostly due to the recurrence of the complications or aesthetic needs.

## 4. Discussion

### 4.1. Safety Assessment of Motiva SilkSurface Breast Implants

The optimal outcome in breast reconstruction necessitates two cardinal features; the finest aesthetic without morbidity. Although both patients and their surgeons are primarily focused on the surgical outcome in terms of aesthetics, safety issues remain a major concern in the long term. Motiva products have been introduced to the market for more than ten years and are claimed to be cell-friendly and safe, as well as ergonomic. The third generation Motiva implants, Ergonomix SilkSurface Silicone Breast Implants, has attracted great attention due to the utilization of low viscosity, highly elastic 100% silicone gel with special rheological properties that would mimic the natural shape and dynamics of the human breast tissue. However, complications related to these implants have not yet been extensively investigated.

Possible postoperative complications of breast augmentation include capsular contracture, hematoma, seroma, chest pain, infection, asymmetry, implant displacement, and implant rupture, among others. The studies included in this manuscript reported a variety of complications again at varying rates, ranging from 0.3% to 68.6%, with a median complication rate of 8.7%. The divergence in reported rates can be justified by different follow-up periods and the surgical techniques in each study.

The TrueMonobloc technology in Ergonomix SilkSurface Silicone Breast Implants discards a gap formation the patch and the shell, forming a single uniform structure with enhanced tensile strength and thereby higher resistance to rupture [22]. The robustness of the shell has been reported to exceed standard regulatory standards and explains the existence of only one single prosthetic rupture complication in the patient series reported in this manuscript.

Coating of the breast implants mainly through glycoprotein deposition due to foreign body reaction is expected and does not typically cause chronic inflammation. However, in certain cases, this reaction can be convoluted by dysregulated immune response and infiltration of the thin fibrous layer around the implants with immune cells. Subclinical bacterial infection and micro degradation of the silicon or silicon itself have been hypothesized to aggravate this response, which can also result in capsular contracture in later stages [12]. Pain and discomfort in patients with capsular fibrosis or, in severe cases, deformation of the breast tissue as well as the implants would increase the need for revision surgery.

Third and fourth-generation breast implants were developed to minimize the incidence of capsule contraction through enhanced silicone cohesion. Elimination of the silicone leakage would prevent an excessive foreign body reaction and reduce the capsular contraction as a post-implantation complication [26]. To hinder the same complication, refinement of the surface topography was targeted in the sixth-generation breast implants [27].

Current research suggests the superiority of implants with textured surfaces over smooth surfaced devices with regard to the reduced incidence of postoperative capsular contracture [27,28]. It has been hypothesized that (1) the textured surfaces interrupt the planar arrangement of fibroblasts [29], (2) the infiltration of the breast tissue into the implants inhibits synovial chemotaxis [19], and (3) oscillatory interstitial fluid stress through micromotion of the implants alters fibroblast activity [30]. However, the unique topography of textured surfaces is suggested to pose a higher risk of breast implant-associated-anaplastic large cell lymphoma (BIA-ALCL) by easing bacterial biofilm formation and intensifying the chronic inflammatory response. Chronically triggered innate immunity and induced T-cell proliferation are thought to account for the malignant transformation in genetically susceptible individuals [31]. Considering the pitfalls of previous surface architectures, a shift to smooth or nano-surface implants appears inevitable.

The Motiva products have been gradually updated utilizing trademark technologies since its introduction, yet the SilkSurface technology remains the “fingerprint” of the product, distinguishing this product from its contenders. The outer shell topography of Motiva is created by 3D-inverted negative imprinting technology on the polydimethylsiloxane (PDMS) material without the use of foreign particles, unlike other implants that use sugar or salt crystal projection to create aggressive textures. The one-of-a-kind fine surface in Motiva implants is suggested to embody extreme delicacy with no loose particles and have 49,000 contact points of 16 μm depth per cm^2^ [32].

Characterizing the physical properties of the implant surface is the key to understanding how surface texture affects tissue response to breast implants. SilkSurface surface technology is used to enhance the biocompatibility of Motiva breast implants by reducing the tissue ingrowth of the implant, optimizing surface adsorption, and avoiding the release of particle fragments.

The extent of tissue ingrowth of the prosthesis is influenced by the surface texture of the implant. Smooth or nanotextured implants, like the Motiva implants, have a smooth and irregular microstructure with no pores, which lessens the number of sites for tissue ingrowth and limits tissue adhesion to the implant [33]. Motiva implants have a layered micro/nanotopography on their surface, which has been shown to affect cell attachment, proliferation, migration, and differentiation in various cell types and matrices [34]. Atlan et al. evaluated the surface texture of 12 different breast implant devices and found that smooth/nanotextured implants had the lowest ingrowth tendency compared to textured tissue [33].

The Motiva prostheses have been described as less invasive than traditional textured prostheses and more resistant to capsular contracture than smooth prostheses, as their silk surface is designed to minimize the foreign body reaction [16]. It has also been demonstrated that nanotextured breast implants may reduce bacterial growth in infected biofilms [35]. In addition, less foreign body reaction with higher levels of immunosuppressive FOXP3+ regulatory T-cell have been observed with the implants with an average roughness of 4 µm both in animals and humans [36].

Despite the presence of individual data sets mostly from single centers and accompanying evidence from translational research, the validation of the superiority of the silk surface breast implant necessitates multicenter clinical trials. In addition, in an era where textured breast implants are being gradually withdrawn from the market, there is still a need and potential for clinical studies comparing the characteristics and surgical outcomes of smooth surface and silk surface implants, especially concerning capsular contracture incidence.

Of the 13 clinical studies included in this study, only three compared the Motiva SilkSurface prostheses with others regarding postoperative complications. In this study, the incidence of capsular contracture after breast augmentation with the Motiva SilkSurface implants ranged from 0–2.06% with a median rate of 0.54%, which is lower than the capsular contracture rate of 3.6% reported by Namnoum for both smooth and textured implants [37]. Similarly, it was lower than what was reported in a ten-year follow-up study including smooth and textured implants [38], where the authors noted a sobering capsular contracture rate of 18.9–28.7%. Nevertheless, due to the limited number of studies with relatively short median follow-ups, we are still far away from reaching an absolute conclusion.

In addition, there are no reported cases of BIA-ALCL associated with Motiva implants. However, this does not specifically substantiate the freedom from BIA-ALCL following breast reconstruction with these implants as BIA-ALCL usually occurs 7–10 years post-surgery. There is only one clinical study in the literature with a follow-up of six years but again with a limited number of patients, and the FDA has reported a single case of BIA-ALCL in a patient with smooth breast implants [6].

With a roughness of 3.18 μm, Motiva SilkSurface breast implants are classified as smooth products according to the International Organization for Standardization (ISO) 14,607 [39]. Considering the tendency of smooth surfaced products to displacement, there might be reservations against these implants in the clinical setting. In the current study, we observed an overall displacement rate of 0.44%. An expert consensus on the application of Motiva SilkSurface prostheses suggests limited dissection for pocket creation to facilitate an optimal fitting of the device and limit potential expansion and displacement [40].

### 4.2. Research Status of Motiva Silksurface Breast Implants

Overall, clinical studies on Motiva SilkSurface breast implants are still limited. This may be partly due to its limited availability in different markets. For instance, the Korean Ministry of Food and Drug Safety (KMFDS) in South Korea approved these implants first in 2016 [17].

Similarly, its clinical investigation by FDA continues today, hence these implants are not still commercially available in the United States [21,41]. A possible reason for its limited adoption might also be the high cost of Motiva implants, which could influence the patient’s choice. According to a study by Moon et al., the mean cost of surgery using Motiva Ergonomix SilkSurface was 8450.02 USD, which was significantly higher when compared to the other six prostheses [15].

As far as the origin of studies in the current literature is concerned, Europe accounts for more than half of the studies on Motiva implants so far. Therefore, the current data that is available cannot be used to make implications on all patients without considering ethnicity-related variables such as skin elasticity, wound healing, and divergence in the immune response. More clinical data on the use of Motiva SilkSurface implant for breast augmentation from Asia, the Americas, Oceania, and Africa would definitely be propitious for optimal patient selection and prevention of potential complications in the future.

For the sake of completeness, it is also crucial to mention the potential limitation of the studies that are included in this study. Most of the current studies are retrospective in nature and prone to selection and information bias. Additionally, the credibility of the studies and their findings are directly affected by the number of patients included and their limited follow-up periods. In addition, most of the studies are designed as single-arm and lack control groups with other prostheses being implanted by the same surgeons with comparable techniques. With different surgical approaches utilized and designs of the studies that are included in this manuscript, it is difficult to conclude exact complication rates and their etiology. Different assessment tools (MRI, ultrasound, or patient report), different surgical incisions, and pocket selection as well as patient characteristics could mask appropriate comparison of the outcome. Additionally, a main limitation of this analysis represents the limited accessibility to standardized high-quality studies with an adequate follow-up time.

It is also worth mentioning that the results of manufacturer-sponsored studies should be interpreted with caution [19,42]. Eight of the thirteen (62%) studies included in this manuscript, either had funding from device manufacturers or one or more authors had a financial relationship with manufacturing companies. Interestingly, a study supported by Establishment Labs showed the lowest complication rate in the largest patient group (*n* = 2502) [25].

Considering the scarcity of the literature on Motiva SilkSurface implants, more prospective, multicenter case-control studies are needed to assess and validate the safety of these implants. National breast implant registries which have been initially established due to concerns about breast implant illness (BII) and BIA-ALCL are and will be the most reliable sources of information on patients with these implants [43]. In addition, despite limited adoption due to patient privacy concerns, the Qid technology could theoretically improve adherence in follow-up, and provide non-biased, standardized data in the near future.

## 5. Conclusions

Although the majority of the studies in the current literature suggest the distinction of the Motiva SilkSurface breast implants in terms of postoperative complications and capsular contracture, its safety and feasibility need to be elucidated with well-designed, large-scale, multicenter, prospective case-control studies with longer follow-ups.

## Figures and Tables

**Figure 1 jcm-12-01881-f001:**
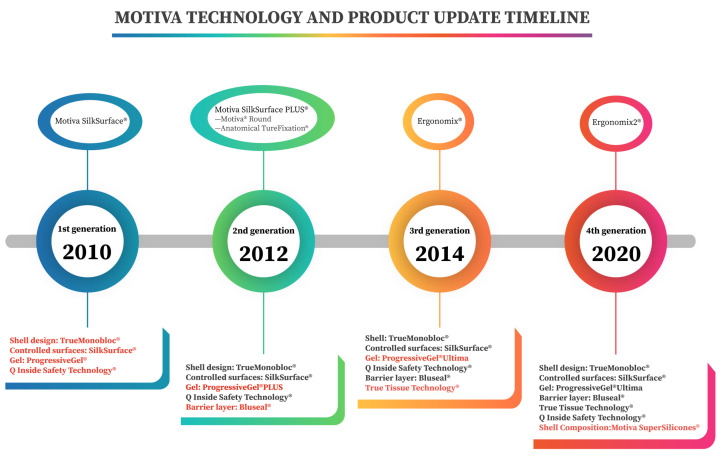
Timeline for Motiva technology and product update.

**Figure 2 jcm-12-01881-f002:**
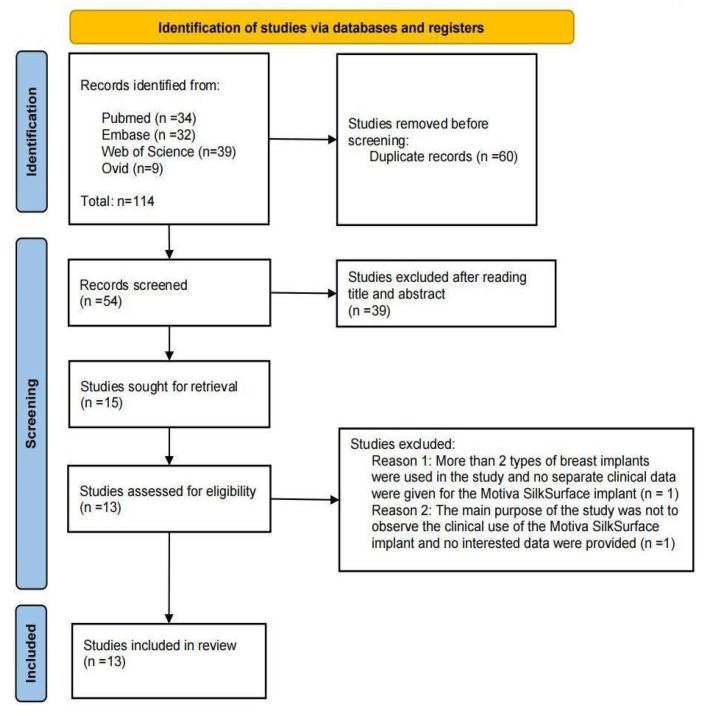
Flow diagram of the literature search and selection (PRISMA compliant flow chart).

**Figure 3 jcm-12-01881-f003:**
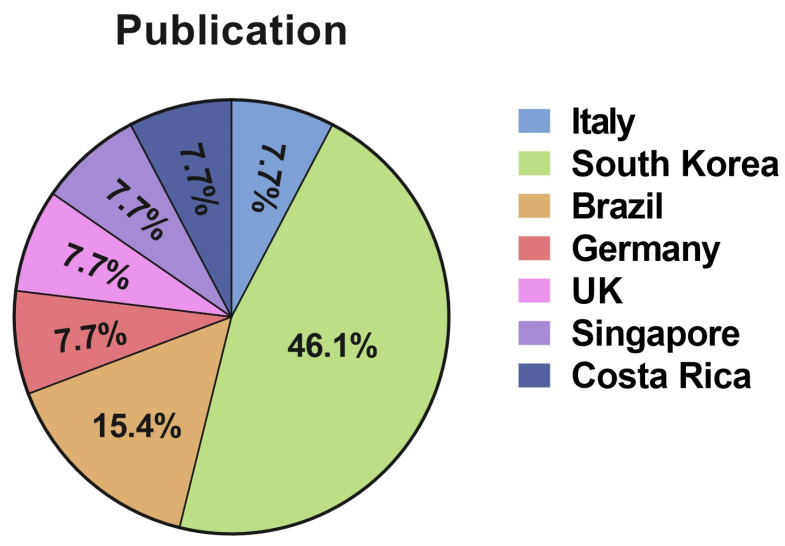
Country distribution of selected Motiva studies.

**Figure 4 jcm-12-01881-f004:**
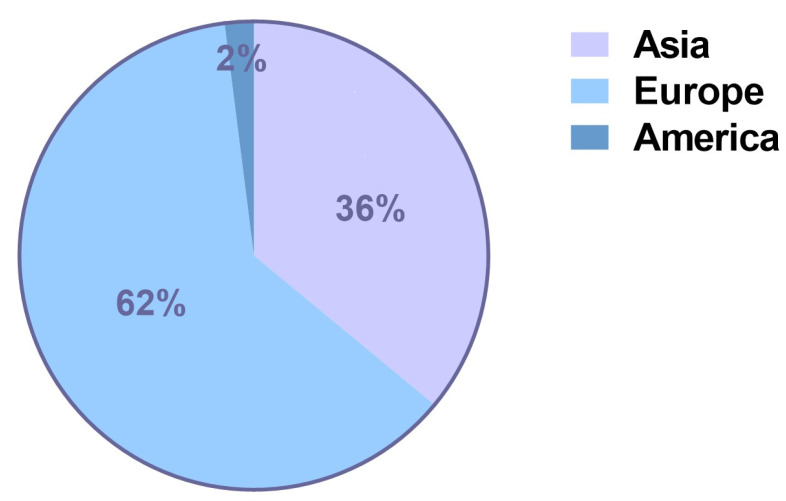
Continent distribution of the patient enrolled in the selected Motiva studies.

**Figure 5 jcm-12-01881-f005:**
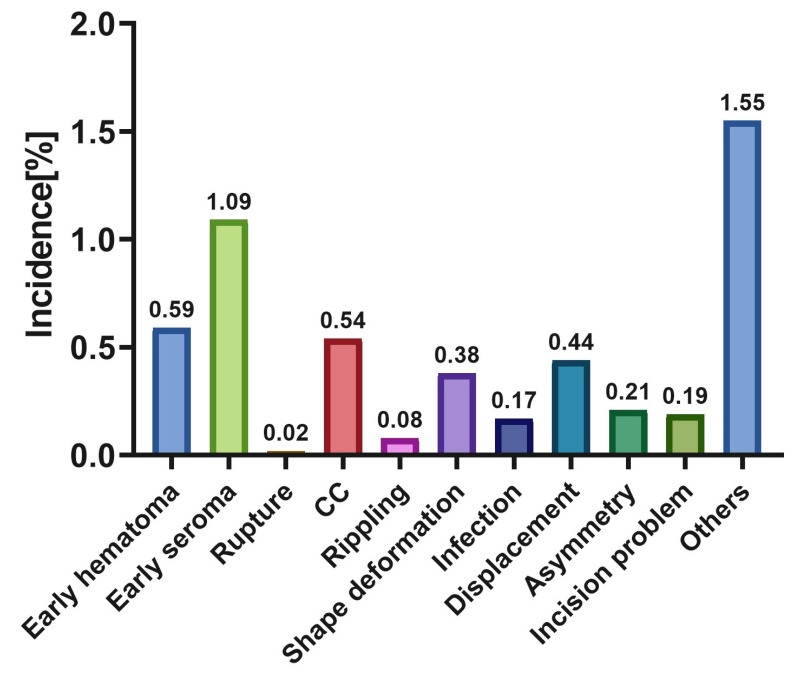
The total incidence of complications.

**Figure 6 jcm-12-01881-f006:**
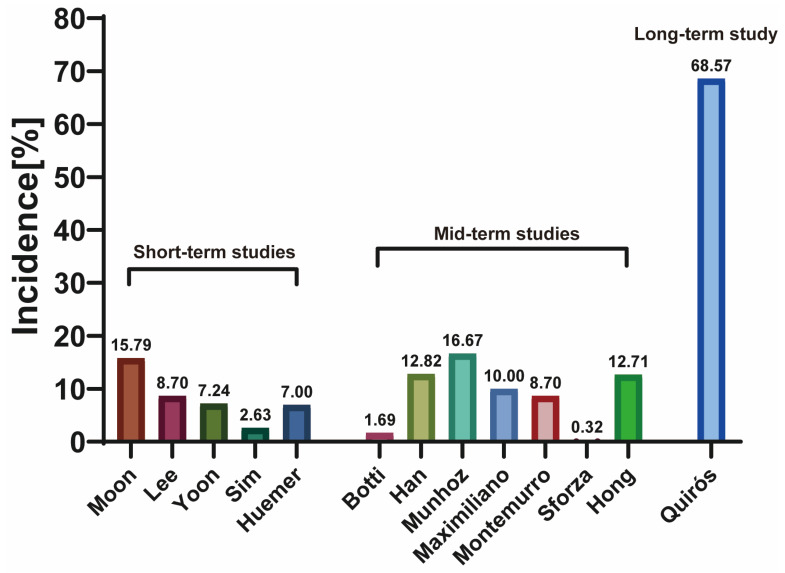
Complication rate of each study in %.

**Table 1 jcm-12-01881-t001:** The basic characteristics of the included literature and patients’ baseline information.

NO	First Author	Year	Journal	Country	Time Period	Number of Patient	Number of Implant	Research Design	FU (Month)	Age	Height (cm)	Weight (kg)	BMI	Disclosures	Funding
1	Giovanni Botti [12]	2022	ASJ	Italy	04.2014–10.2018	356	712	Retrospective	35.9	34.1	NA	NA	NA	Dr. Botti is an educator in the MotivaEdge (Motiva, Houston, TX, USA)	none
2	Sanghyuk Han [13]	2021	Medicina	Korea	01.2017–08.2021	312	624	Retrospective	12.68 ± 0.58	34.19 ± 8.62	163.58 ± 5.09	52.29 ± 5.78	NA	none	none
3	Alexandre Mendonça Munhoz [14]	2021	ASJ	Brazil	06.2017–02.2019	42	84	Retrospective	18	34.6	NA	NA	18.8	Dr.Munhoz serves as a consultant/board member for Establishment Labs, Holdings, Inc. (Alajuela, Costa Rica) and has shares of stocks in the company	none
4	Dong Seung Moon [15]	2021	Journal of Plastic Surgery and Hand Surgery	Korea	09.2017–04.2019	76	152	Retrospective	4.2 ± 3.88	35.84 ± 8.60	163.67 ± 5.12	52.45 ± 5.72	NA	none	none
5	Paolo Montemurro [16]	2021	ASJ	Singapore	07.2016–03.2019	161	322	Retrospective	24.3	30.8	NA	NA	20.36	Dr. Montemurro served as an independent speaker for Motiva (Establishment Labs, Alajuela, Costa Rica)	none
6	Sangdal Lee [17]	2021	ASJ open	Korea	09.2017–12.2020	69	138	Retrospective	9.75 ± 9.21	34.2 ± 8.2	163.5 ± 5.1	51.6 ± 5.5	19.3 ± 1.8	Dr.Lee is an investigator, speaker, and consultant for Motiva Korea Ltd.	none
7	Pa Hong [19]	2021	APS	Korea	09.2016–08.2020	873	1746	Retrospective	18.50 ± 5.88	32.18 ± 6.88	165.67 ± 15.23	49.37 ± 4.16	NA	none	none
8	Seanhyuck Yoon [20]	2020	PRS open	Korea	01.2017–03.2018	152	304	Retrospective	7.17 ± 5.36	36.67 ± 7.77	161.75 ± 5.37	50.58 ± 5.34	NA	none	This study was sponsored by the HansBiomed Co. Ltd.
9	João Maximiliano [21]	2020	ASJ	Brazil	06.2017–04.2019	30	60	Prospective	18	33	NA	NA	21.1	Dr Munhoz serves as a consultant/board member for Establishment Labs, Holdings, Inc; and has shares of stocks in the company	none
10	Manuel Chacón Quirós [22]	2019	ASJ	Costa Rica	09.2010–12.2010	35	70	Prospective	72	31.5	NA	55.8	21.9	Dr.Manuel Chacón Quirós and Dr.Manuel Chacón Bolaños are relatives of the CEO of Establishment Labs.	this study was funded by Establishment Labs Holdings Inc. (New York, NY)
11	Hyung-Bo Sim [23]	2019	ASJ	Korea	06.2015–05.2018	76	152	Prospective	12	27.7	165.2	53.4	19.5	none	none
12	Georg M. Huemer [24]	2018	PRS	Germany	2014–2017	100	200	Retrospective	minimum of 6	32.8	NA	NA	20.6	none	none
13	Marcos Sforza [25]	2018	ASJ	UK	04.2013–04.2016	2502	5004	Retrospective	23.03	28.2 ± 10.98	NA	NA	NA	Dr. Sforza serves as coordinator of the Medical Advisory Board, has a consulting agreement with Establishment Labs Holdings, Inc	This article was supported by Establishment Labs (Alajuela, Costa Rica)

**Table 2 jcm-12-01881-t002:** The characteristics of implants.

NO	First Author	Implant	Generation	Volume (CC)					Profile				Projection			
				≤245CC	250–295CC	300–345CC	350–395CC	≥400CC	Ultrahigh	High	Medium	Low	Corsé	Full	Demi	Mini
1	Giovanni Botti [12]	Motiva Ergonomix^®^	3	mean = 240cc					NA				NA			
2	Sanghyuk Han [13]	Motiva Ergonomix^®^	3	6	32	54	34	30	360	192	72	0	NA			
3	Alexandre Mendonça Munhoz [14]	Motiva Ergonomix^®^	3	mean = 255cc					NA				0	84	0	0
4	Dong Seung Moon [15]	Motiva Ergonomix^®^	3	6	31	55	36	24	0	130	22	0	NA			
5	Paolo Montemurro [16]	Motiva Ergonomix^®^/Motiva SilkSurface PLUS^®^(Round)	2 or 3	mean = 341.82cc					NA				NA			
6	Sangdal Lee [17]	Motiva Ergonomix^®^	3	3	55	46	22	12	0	124	14	0	NA			
7	Pa Hong [19]	Motiva Ergonomix^®^	3	30	228	972	420	96	NA				NA			
8	Seanhyuck Yoon [20]	Motiva Ergonomix^®^	3	29	91	162	22		NA				NA			
9	João Maximiliano [21]	Motiva Ergonomix^®^	3	mean = 265cc					0	60	0	0	0	60	0	0
10	Manuel Chacón Quirós [22]	Motiva SilkSurface^®^	1	mean = 326.70					NA				NA			
11	Hyung-Bo Sim [23]	Motiva Ergonomix^®^	3	29	81	40	2	0	NA				0	3	142	7
12	Georg M. Huemer [24]	Motiva Ergonomix^®^	3	mean = 368cc					NA				2	130	68	0
13	Marcos Sforza [25]	Motiva SilkSurface PLUS^®^	2	NA					NA				NA			

**Table 3 jcm-12-01881-t003:** The surgical techniques of included studies.

NO	First Author	BA type		Incision				Pocket					Number of Surgeons	Pocket Irrigation	Others
		Primary	Secondary	Axillary	IMF	Peri-Areolar	Others	Sub- Pectoral	Sub- Glandular	Sub-Fascial	Dual-plane	Others			
1	Giovanni Botti [12]	282	74	54	196	107	0	28	0	0	328	0	NA	×	
2	Sanghyuk Han [13]	312	0	268	28	16	0	253	59	0	0	0	4	H2O2 solution, betadine	montelukast sodium was used
3	Alexandre Mendonça Munhoz [14]	23	19	42	0	0	0	0	0	42	0	0	1	antibiotic solution	AFG
4	Dong Seung Moon [15]	NA	NA	65	7	4	0	0	0	0	76	0	1	H2O2 solution, betadine	montelukast sodium was used
5	Paolo Montemurro [16]	161	0	0	161	0	0	0	13	0	148	0	1	antibiotic solution	
6	Sangdal Lee [17]	69	0	68	0	1	0	69	0	0	0	0	NA	H2O2 solution, betadine	
7	Pa Hong [19]	873	0	0	873	0	0	0	0	0	873	0	NA	Betadine Triple Antibiotic solution	
8	Seanhyuck Yoon [20]	130	22	0	147	4	1	0	0	0	152	0	2	NA	
9	João Maximiliano [21]	30	0	25	5	0	0	0	0	30	0	0	1	NA	AFG
10	Manuel Chacón Quirós [22]	35	0	1	34	0	0	24	5	5	1	0	NA	×	
11	Hyung-Bo Sim [23]	76	0	76	0	0	0	0	0	76	0	0	NA	povidone-iodine, gentamicin, 10% tranexamic acid, and normal saline	AFG
12	Georg M. Huemer [24]	100	0	0	100	0	0	0	0	0	100	0	1	NA	
13	Marcos Sforza [25]	2126	376	0	most	NA	NA	NA	NA	0	most	NA	16	×	

**Table 4 jcm-12-01881-t004:** The complications and reoperation of included studies.

NO	First Author	NO of Patient	Reoperations	Reoperation Rate	Complications	Incidence	Complications										
							Early Hematoma	Early Seroma	Rupture	CC	Rippling	Shape Deformation	Infection	Displacement	Asymmetry	Incision Problem	Others
1	Giovanni Botti [12]	356	NA	NA	6	1.69%	3	0	0	2	0	0	0	1	0	0	0
2	Sanghyuk Han [13]	312	NA	NA	40	12.82%	4	20	0	0	0	12	0	0	0	0	Stretch deformities with skin excess: 4
3	Alexandre Mendonça Munhoz [14]	42	1	2.38%	7	16.67%	1	0	0	1	0	0	0	0	0	2	subcutaneous banding in the axilla: 3
4	Dong Seung Moon [15]	76	NA	NA	12	15.79%	1	4	0	0	0	4	0	1	0	0	thickened capsule:1,others:1
5	Paolo Montemurro [16]	161	NA	NA	14	8.70%	0	0	0	2	0	0	0	12	0	0	0
6	Sangdal Lee [17]	69	NA	NA	6	8.70%	0	2	0	1	0	2	0	0	0	0	foreign body sensation:1
7	Pa Hong [19]	873	NA	NA	111	12.71%	18	24	0	18	3	0	6	0	9	0	Dissatisfaction with shape:17,Dissatisfaction with size:16
8	Seanhyuck Yoon [20]	152	NA	NA	11	7.24%	0	1	0	2	1	0	1	3	1	0	Dissatisfaction with size:1,Psychological distress:1
9	João Maximiliano [21]	30	1	3.33%	3	10.00%	0	0	0	0	0	0	0	0	0	1	subcutaneous banding in the axilla: 2
10	Manuel Chacón Quirós [22]	35	3	8.57%	24	68.57%	0	0	0	0	0	0	0	0	0	0	Changes in nipple sensitivity:3,Pain:2,Ptosis:17,Twinges:2
11	Hyung-Bo Sim [23]	76	0	0.00%	2	2.63%	0	0	0	0	0	0	0	0	0	0	Contour visibility:2
12	Georg M. Huemer [24]	100	7	7.00%	7	7.00%	1	0	1	0	0	0	0	4	0	0	Dissatisfaction with size:1
13	Marcos Sforza [25]	2502	NA	NA	8	0.32%	0	1	0	0	0	0	1	0	0	6	0
Total		4784	——	——	251	5.25%	28	52	1	26	4	18	8	21	10	9	74

## Data Availability

All included data will be available on request to the corresponding author.

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
