# Peer review of "Complication Rates after Breast Surgery with the Motiva Smooth Silk Surface Silicone Gel Implants—A Systematic Review and Meta-Analysis"

_jcm, 2023, doi:10.3390/jcm12051881_

Round 1

Reviewer 1 Report

Dear authors, I woud like to cheer you for your interesting review. Indeed, breast implant world is rapidly expanding and new concerns as ALCL disease underlined the importance to improve our knowledge in this field. I have only few suggestions for you:

- 3.3 complications section, page 7, line 161: “Of the 4784 patients who received breast augmentation with Motiva SilkSurface ...”. It is not clear here if all the patients inclulded in the study underwent aesthetic breast aumentation or part of them underwent breast reconstruction. Indeed, at the beginning of the paper, authors state they plan to include both aesthetic augmentations and breast reconstructions. As complication rates variate consistently between the two groups, it would be useful if the authors specify more clearly this detail.

- 4. Discussion section, page 8, line 192-193: “The divergence in reported rates can be justified by different follow-192 up periods and the surgical techniques in each study ”. If both patients that underwent aestetic breast augmentation and cases of breast reconstruction were included in the study, divergence in complication rates may be due to their higher incidence in post-mastectomy cases. Please specify it.

Author Response

Dear authors, I woud like to cheer you for your interesting review. Indeed, breast implant world is rapidly expanding and new concerns as ALCL disease underlined the importance to improve our knowledge in this field. I have only few suggestions for you:

- 3.3 complications section, page 7, line 161: “Of the 4784 patients who received breast augmentation with Motiva SilkSurface ...”. It is not clear here if all the patients inclulded in the study underwent aesthetic breast aumentation or part of them underwent breast reconstruction. Indeed, at the beginning of the paper, authors state they plan to include both aesthetic augmentations and breast reconstructions. As complication rates variate consistently between the two groups, it would be useful if the authors specify more clearly this detail.

Thank you for reviewing our manuscript and making this point. In fact, we decided further on to include only breast augmentation, so "Breast reconstruction" was removed from the inclusion criteria.

- 4. Discussion section, page 8, line 192-193: “The divergence in reported rates can be justified by different follow-192 up periods and the surgical techniques in each study ”. If both patients that underwent aestetic breast augmentation and cases of breast reconstruction were included in the study, divergence in complication rates may be due to their higher incidence in post-mastectomy cases. Please specify it.

Thank you very much. As mentioned above, we have removed "Breast reconstruction" from the inclusion criteria. Initially, we planned to include "Breast reconstruction" in the meta-analysis,  but decided against it later on.

Reviewer 2 Report

Thank you for requesting  to provide a review of this article, which has a subject of high interest. 

   The main purpose of the analysis was to investigate the outcome and complication rates of Motiva SilkSurface breast implants in clinical use, to provide an evidence-based safety assessment.

   The main question adressed in the research was whether anaplastic large cell lymphoma (BIA-ALCL) might be associated with Motiva SilkSurface breast implants and which are the most common complications related to this kind of surgery.

   The study was performed in accordance with the Cochrane handbook and PRISMA guidelines and randomized controlled trials, cross-sectional and cohort studies on Motiva implants were included. From a total of 4784 patients who underwent breast augmentation with Motiva SilkSurface, a total of 250 complications were observed, the most common complication being early seroma, followed by early hematoma. The topic is original and relevant in the field and brings usefull knowledge regarding the subject. A comprehensive search strategy was used. The review methodology was comprehensive with screening and data extraction. When it comes to the methodology used, no specific improvements should be considered from my point of view.

   The conclusions are consistent with the evidence and the arguments presented, and they adress properly to the main question which conducted the analysis.

   The references are appropriate and well suited for this kind of study. 

    Regarding the figures and pictures used in the article, they provide suitable information about the cases and show significant statistical references. They are also well understandable and the information is easy to be followed. There are no other comments required about these items, from my point of view.

  Regarding the structure and accuracy of the phrases, the manuscript has well structured information, with supported evidence and well structured phrases.

   The manuscript is original and well defined. The results provide an advance in current knowledge. The results are being interpreted appropriately and are significant, as well as the conclusions.

  The article is written in an appropriate way. 

  The study is correctly designed and the analysis is being performed at high standards, so the data are robust enough to draw the conclusion. 

   Surely the paper will attract a wide readership. 

   The English language is appropriate and well understandable.

   There are only 3 corrections to be added in the lines below, but the article should be definetelly published after the corrections are made: 

Line 75: „,” after „use”

Line 100: which, not „those”

Line 111: „;” instead of „.” after „surgeons” 

Author Response

Thank you for requesting  to provide a review of this article, which has a subject of high interest. 

   The main purpose of the analysis was to investigate the outcome and complication rates of Motiva SilkSurface breast implants in clinical use, to provide an evidence-based safety assessment.

   The main question adressed in the research was whether anaplastic large cell lymphoma (BIA-ALCL) might be associated with Motiva SilkSurface breast implants and which are the most common complications related to this kind of surgery.

   The study was performed in accordance with the Cochrane handbook and PRISMA guidelines and randomized controlled trials, cross-sectional and cohort studies on Motiva implants were included. From a total of 4784 patients who underwent breast augmentation with Motiva SilkSurface, a total of 250 complications were observed, the most common complication being early seroma, followed by early hematoma. The topic is original and relevant in the field and brings usefull knowledge regarding the subject. A comprehensive search strategy was used. The review methodology was comprehensive with screening and data extraction. When it comes to the methodology used, no specific improvements should be considered from my point of view.

   The conclusions are consistent with the evidence and the arguments presented, and they adress properly to the main question which conducted the analysis.

   The references are appropriate and well suited for this kind of study. 

    Regarding the figures and pictures used in the article, they provide suitable information about the cases and show significant statistical references. They are also well understandable and the information is easy to be followed. There are no other comments required about these items, from my point of view.

  Regarding the structure and accuracy of the phrases, the manuscript has well structured information, with supported evidence and well structured phrases.

   The manuscript is original and well defined. The results provide an advance in current knowledge. The results are being interpreted appropriately and are significant, as well as the conclusions.

  The article is written in an appropriate way. 

  The study is correctly designed and the analysis is being performed at high standards, so the data are robust enough to draw the conclusion. 

   Surely the paper will attract a wide readership. 

   The English language is appropriate and well understandable.

   There are only 3 corrections to be added in the lines below, but the article should be definetelly published after the corrections are made: 

Line 75: „,” after „use”

Line 100: which, not „those”

Line 111: „;” instead of „.” after „surgeons”

We thank you for the review of our manuscript and for the summary of our article.

The three changes were implemented in the manuscript.

Reviewer 3 Report

Dear Authors, 

Thank you for the opportunity to read the manuscript entitled: "Complication Rates after Breast Surgery with the Motiva Smooth Silk Surface Silicone Gel Implants- A Safety Assessment" by Aitzetmüller-Klietz et al. The paper is interesting, in scientific language, but has some flaws. 

1) the type of article - since it is a Review article the type of article should be selected "article" suggests that it is an original article from the authors' research work - and according to the submitted manuscript it is a Review or Meta-analysis or Systemic Review type (please detail the type of article and also indicate this in the title of the article e.g. - A review or - Systemic Review etc. 

2) Introduction section : 

The paper is about implants, and breast surgery in general however modern elements of breast reconstructive surgery are missing e.g. the section on the use of acellular dermal matrix PMID: 36359387. 

This expansion will improve the very sparse references section.

3) References style is not in MDPI Style - needs a complete overhaul. 

4) Why only 13 papers were included in this meta-analysis out of 54 ?As I understand it because of the type of implant ? Only 13 papers compared these implants ? Why only MOTIVA implants ? 

5) Spell check required.

6) You should write under the flow chart that this is a PRISMA compliant flow chart.

Thank you for the opportunity to read - the work in general is good, but needs some corrections, I hope the authors will address my suggestions well.

Best regards,

Author Response

Dear Authors, 

Thank you for the opportunity to read the manuscript entitled: "Complication Rates after Breast Surgery with the Motiva Smooth Silk Surface Silicone Gel Implants- A Safety Assessment" by Aitzetmüller-Klietz et al. The paper is interesting, in scientific language, but has some flaws. 

  • the type of article - since it is a Review article the type of article should be selected "article" suggests that it is an original article from the authors' research work - and according to the submitted manuscript it is a Review or Meta-analysis or Systemic Review type (please detail the type of article and also indicate this in the title of the article e.g. - A review or - Systemic Review etc. 

We thank the reviewer for this insightful comment. According to the fact we analyzed and summarized the papers included, this article represents an original article.

2) Introduction section : 

The paper is about implants, and breast surgery in general however modern elements of breast reconstructive surgery are missing e.g. the section on the use of acellular dermal matrix PMID: 36359387. 

This expansion will improve the very sparse references section.

We thank the reviewer for his concern. The referenced article is about Hidradenitis suppurativa, which is far beyond the scope of our manuscript (texture of implants).

3) References style is not in MDPI Style - needs a complete overhaul. 

 Thank you for this concern. The reference style was adapted to MDPI Style.

4) Why only 13 papers were included in this meta-analysis out of 54 ?As I understand it because of the type of implant ? Only 13 papers compared these implants ? Why only MOTIVA implants ? 

Motiva implants represent the first breast implants with nano-texturing available on the market and are therefore characterized by the largest number of long-term data. For alternative implants with nano-texturing no adequate studies exist yet.

Why only 13 papers were included in this meta-analysis out of 54?

The remaining papers, which were not used, did not meet the inclusion criteria for the study. This is illustrated by the decision tree.

5) Spell check required.

 Spell check done.

6) You should write under the flow chart that this is a PRISMA compliant flow chart.

 Thank you for pointing this out. We added that this is a PRISMA compliant flow chart.

Thank you for the opportunity to read - the work in general is good, but needs some corrections, I hope the authors will address my suggestions well.

Best regards,

Round 2

Reviewer 3 Report

Dear Authors, 

Thank you for your responses however I believe : 

1) This article does not present the type of original article , but it is a literature review also the type of article here is review. The title should indicate the type of study as in original articles it can be "retrospective comparative study" or "Prospective comparative cohort study" etc. Same here - this is a literature review you can treat it as "meta-analysis" or "systemic review". In your response to another reviewer you used the word "meta-analysis". 

2)not true - PMID: 36359387 - this is a review article on ADM and not in the treatment of HS, where there is a piece on ADM in applications to breast plastic surgery (also there is a piece on HS but this is currently the most extensive review on ADM - publication year 2022) . In my opinion, the description of current methods that improve surgical treatment should be included in your paper even a short sentence about the use of acellular dermal matrix. E.g., "In breast plastic surgery and especially in oncoplastic breast reconstructive surgery, acellular dermal matrices are used, which significantly improve surgical conditions by also promoting healing and forming aesthetic scars."

I think the authors will understand my suggestions. 

With best regards. 

Author Response

Dear Authors, 

Thank you for your responses however I believe : 

This article does not present the type of original article , but it is a literature review also the type of article here is review. The title should indicate the type of study as in original articles it can be "retrospective comparative study" or "Prospective comparative cohort study" etc. Same here - this is a literature review you can treat it as "meta-analysis" or "systemic review". In your response to another reviewer you used the word "meta-analysis". 

We thank the reviewer for this suggestion. We added the article type (systematic review and meta-analysis) to the title.

not true - PMID: 36359387 - this is a review article on ADM and not in the treatment of HS, where there is a piece on ADM in applications to breast plastic surgery (also there is a piece on HS but this is currently the most extensive review on ADM - publication year 2022) . In my opinion, the description of current methods that improve surgical treatment should be included in your paper even a short sentence about the use of acellular dermal matrix. E.g., "In breast plastic surgery and especially in oncoplastic breast reconstructive surgery, acellular dermal matrices are used, which significantly improve surgical conditions by also promoting healing and forming aesthetic scars."

 We thank the reviewer for this comment. The given study was implemented in the introduction as suggested.

I think the authors will understand my suggestions. 

With best regards. 
